# Jasmonic Acid Signaling Pathway in Plants

**DOI:** 10.3390/ijms20102479

**Published:** 2019-05-20

**Authors:** Jingjun Ruan, Yuexia Zhou, Meiliang Zhou, Jun Yan, Muhammad Khurshid, Wenfeng Weng, Jianping Cheng, Kaixuan Zhang

**Affiliations:** 1College of Agriculture, Guizhou University, Guiyang 550025, China; jjruan@gzu.edu.cn (J.R.); zhouyuexiagui@hotmail.com (Y.Z.); wenfengweng@hotmail.com (W.W.); jpcheng@gzu.edu.cn (J.C.); 2Institute of Crop Sciences, Chinese Academy of Agricultural Sciences, Beijing 100081, China; zhoumeiliang@caas.cn (M.Z.); khurshid.ibb@pu.edu.pk (M.K.); 3Schools of Pharmacy and Bioengineering, Chengdu University, Chengdu 610106, China; yanjun62@cdu.edu.cn; 4Institute of Biochemistry and Biotechnology, University of the Punjab, Lahore 54590, Pakistan

**Keywords:** jasmonic acid, signaling pathway, environmental response, biological function

## Abstract

Jasmonic acid (JA) and its precursors and dervatives, referred as jasmonates (JAs) are important molecules in the regulation of many physiological processes in plant growth and development, and especially the mediation of plant responses to biotic and abiotic stresses. JAs biosynthesis, perception, transport, signal transduction and action have been extensively investigated. In this review, we will discuss the initiation of JA signaling with a focus on environmental signal perception and transduction, JA biosynthesis and metabolism, transport of signaling molecules (local transmission, vascular bundle transmission, and airborne transportation), and biological function (JA signal receptors, regulated transcription factors, and biological processes involved).

## 1. Introduction

Plants undergo many physiological changes to cope with biotic and abiotic stress. The survival of plants mainly depends on their ability to adapt in a varying environment through signaling networks [1]. These networks establish connections between the environmental signals and cell responses [2]. Plant hormones play major roles in the establishment of signaling networks to regulate plant growth and stress-related responses. Jasmonic acid (3-oxo-2-2′-*cis*-pentenyl-cyclopentane-1-acetic acid, abbreviated as JA) is an endogenous growth-regulating substance found in higher plants. JA and its methyl ester (MeJA) and isoleucine conjugate (JA-Ile) are derivatives of a class of fatty acids and are collectively known as jasmonates (JAs). Initially identified as a stress-related hormone, JAs are also involved in the regulation of important growth and developmental processes [3,4]. For example, JAs can induce stomatal opening, inhibit Rubisco biosynthesis, and affect the uptake of nitrogen and phosphorus and the transport of organic matter such as glucose. In particular, as a signaling molecule, JAs can effectively mediate responses against environmental stresses by inducing a series of genes expression [5]. JAs and salicylic acid (SA)-mediated signaling pathways are mainly related to plant resistance, prompting plant responses to external damage (mechanical, herbivore, and insect damage) and pathogen infection, thereby inducing resistance gene expression. In this review, the initiation, transmission, and biological functions of jasmonic acid signaling are introduced from the point of view of environmental signal molecules.

## 2. Initiation of Jasmonic Acid Signaling

### 2.1. Signal Perception and Transduction

In the last decades, a large number of studies have been conducted on how biotic and abiotic stress signals are perceived by plants and the biosynthesis of JA is initiated. In tomato (*Lycopersicon esculentum*), Pearce et al. in 1991 found a systemin that responded to mechanical damage such as insect damage [6]. Systemin is a polypeptide signal molecule consisting of 18 amino acids derived from a precursor protein consisting of 200 amino acids: prosystemin [7]. After the tomato is mechanically damaged, the prosystemin is hydrolyzed into a systemin, which can be transported to other cells via the apoplast and combined with the cell surface receptor SR160 (a protein rich in leucine repeat units) to finally activate the JA signaling pathway [8,9]. In addition to the traumatic signals, oligosaccharide signals induced by pathogens and fungal elicitors were also found in tomatoes, which ultimately activate the JA signaling pathway. It is speculated that the mechanism of action of oligosaccharides has a similar pathway to that of the systemin, but the specific mechanism of induction is still unclear [10,11].

Later, a polypeptide having the same function as a systemin, AtPEP1, consisting of 23 amino acids, was also found in *Arabidopsis thaliana*. Similar to the production of systemin, mechanical damage or pathogen infection promotes the hydrolysis of the precursor protein PROPEP1 (consisting of 92 amino acids) to AtPEP1, which binds to the receptor PEPR1 (an enzyme rich in leucine repeat units) on the plasma membrane, which ultimately activates the JA signaling pathway [12].

The process by which systemin and AtPEP1 bind to the receptor to activate the JA signaling pathway is complex. It is known that the most important process is to activate phospholipase on the plasma membrane, and then the phospholipase acts on the membrane to release linolenic acid (a precursor of JA synthesis) from the phospholipid [13]. However, the mechanism by which systemin activates phospholipase is unclear. So far, several phospholipases that can be induced by systemin and AtPEP1 have been identified, including PLA2 in tomato and DAD1, DGL, and PLD in *Arabidopsis*, and these phospholipases have similar mechanisms of action [14,15,16].

### 2.2. Synthesis and Metabolism of Jasmonate Compounds

Biosynthesis of JAs has been studied in a variety of monocotyledonous and dicotyledonous plants during the last decades. Most of the work is done in the model plants *Arabidopsis thaliana* and *L. esculentum* (tomato). So far, various enzymes in the JAs synthetic pathway have been identified, and our knowledge of the relationship between the JA synthesis pathway and other metabolic pathways is gradually improving (Figure 1). In *Arabidopsis*, there are three pathways for the synthesis of JAs, including the octadecane pathway starting from α-linolenic acid (18:3) and the hexadecane pathway starting from hexadecatrienoic acid (16:3) [17]. All three pathways require three reaction sites: the chloroplast, peroxisome, and cytoplasm. The synthesis of 12-oxo-phytodienoic acid (12-OPDA) or deoxymethylated vegetable dienic acid (dn-OPDA) from unsaturated fatty acid takes place in the chloroplast, which is then converted to JA in the peroxisome. In the cytoplasm, JA is metabolized into different structures by various chemical reactions, such as MeJA, JA-Ile, *cis*-jasmone (CJ), and 12-hydroxyjasmonic acid (12-OH-JA).

## 3. Transmission of the Jasmonic Acid Signal

The defense response triggered by a traumatic signal can result in a local defense response near the wound and/or a systemic acquired resistance (SAR) at the uninjured site, and/or even induced defense responses from adjacent plants. In these defense responses, short-distance transmission and long-distance transmission of JA signals are involved [18]. With the studies in the area of mechanisms of hormone signaling networks, it has been found that salicylic acid, ethylene, auxin, and other plant hormones interact with JA to regulate plant adaptation to the environment. At present, the understanding of complex regulatory networks and metabolic processes after plants perceive environmental signals is still very limited.

### 3.1. Short-Distance Signal Transmission

In plants, mechanical damage or insect feeding can cause rapid and transient accumulation of JA and JA-Ile at the site of injury, thereby activating the expression of defense genes surrounding the wound and producing a local defense response. In the local defense response, there are two ways of short-distance transmission of the JA signal. First, the systemin produced by the wounding acts as a signaling molecule, which is transmitted to the adjacent site through the apoplast and phloem to activate the JA cascade reaction pathway. Second, JA and JA-Ile induced by systemin act as signals and are transported to adjacent sites for defensive responses [19].

### 3.2. Long-Distance Signal Transmission

So far, it is known that the long-distance transmission of JA signals is via vascular bundle transmission and/or airborne transmission.

#### 3.2.1. Vascular Bundle Transmission

Previously, many researchers believed that systemin functions directly in the long-distance signal transmission and is a mobile signal molecule. However, a series of grafting experiments using tomato jasmonate-insensitive mutant (*jai1*), systemin-insensitive mutant (*spr1*), and the JAs biosynthesis deletion mutants *spr2* and *acx1A* demonstrated that the systemin was not the systemically transmitted signal [19]. After the induction of the synthesis of JA, JA and MeJA are systemically transmitted in plants [20]. Thorpe et al. demonstrated by radioisotopic labeling experiments that MeJA can be transferred to phloem and xylem in vascular bundles [21]. Some work has also shown that JAs are not simply transported along the vascular bundle, but are accompanied by resynthesis of JAs during transport [20]. The localization of various JA synthetases (such as LOX, AOS, etc.) was also found in the companion cell–sieve element complex (CC-SE) of tomato vascular bundles [22], and the sieve molecules in the phloem have the ability to form the JA precursor OPDA [23]. Recently, Koo et al. [24] found that the systemic JA and JA-Ile caused by injury induction are not all transferred from the injured site, at least part of which is resynthesized and cascading cycles in the uninjured site produce more JA-Ile, which was later confirmed by Larrieu et al. [25].

#### 3.2.2. Airborne Transmission

It was found that the flow rate of the tomato phloem signal is 1–5 cm per hour [26], but the accumulation of JA and JA-Ile can be detected in the whole plant within 15 min after mechanical damage [27]. In the 1990s, ring-cutting experiments demonstrated that although the vascular bundle transmission was blocked, there was still a rapid and strong defense gene expression in the distal leaves [27]. A large number of studies showed that in addition to vascular bundle transmission, there are other long-distance transmission routes for JA signals. Compared with JA, which has difficulty in penetrating the cell membrane without carrier assistance, MeJA easily penetrates the cell membrane and has strong volatility, and thus can be spread by airborne diffusion to distant leaves and adjacent plants [28]. It has been confirmed in a range of plants, such as *Arabidopsis thaliana* [20], *Nicotiana tabacum* [29], *Phaseolus lunatus* [30], and *Artemisia kawakamii* [31], that MeJA can be transmitted by air between damaged and undamaged leaves or between adjacent plants.

## 4. Perception of the Jasmonic Acid Signal and Induction of Response

### 4.1. Jasmonic Acid Receptor

The nuclear transport mechanism of JAs was systematically analyzed by means of molecular genetics, molecular biology, biochemistry, and cell biology. The ABC transporter AtJAT1/AtABCG16 with JAs transport ability was screened by a yeast system [32]. Radioactive isotope uptake experiments and autoradiography experiments showed that AtJAT1/AtABCG16 acts as a high-affinity transporter to regulate the subcellular distribution of JAs [32]. AtJAT1/AtABCG16 is localized on the nuclear and plasma membranes of plant cells and mediates the transport of JAs across the plasma membrane and the bioactive JA-Ile across the inner membrane of the nuclear membrane to activate JA responses at low concentration. When the concentration of JAs is high, the function of the JA transporter on the cytoplasmic membrane is dominant, which reduces the intracellular JA and JA-Ile concentrations to desensitize the JA signal. The JAs signaling pathway is activated in other cells by transporting JA to the apoplast. AtJAT1/AtABCG16 can rapidly regulate the dynamics of JA/JA-Ile in cells, which leads to the quick transport of JA-Ile into the nucleus when the plant is under stress, as well as the quick desensitization of the JA signal to avoid the inhibition of plant growth and development by the defense response (Figure 2).

The understanding of JA receptors has undergone a complex process. In 1994, Feys first found that the *Arabidopsis coronatine insensitive1* (*coi1*) mutant lost all responses to JA [33], and further studies indicated that the *COI1* gene encodes an F-box protein that is a component of E3 ubiquitin ligase [34]. In this case, COI1 associates with the SKP1 protein and Cullin protein to form the SCF-type E3 ubiquitin ligase that is referred to as SCF^COI1^, which targets the repressor proteins for degradation by ubiquitination [34,35]. The discovery of COI1 protein is of great significance for the study of theJA signaling pathway.

It was once thought that COI1 is the receptor for jasmonic acid signaling in cells, until the discovery of a jasmonate Zinc finger Inflorescence Meristem (ZIM)-domain (JAZ) protein family, which gave a new understanding of the jasmonic acid signal transduction pathway. In 2007, three research groups simultaneously found that JAZ proteins act as repressors in the JA signaling pathway [36,37,38]. To date, 13 JAZ proteins have been found in *Arabidopsis*, most of which have two conserved domains, Jas and ZIM [39]. The JAZ protein interacts with COI1 via the Jas domain and interacts with MYC2 via the ZIM domain [40]. Therefore, many researchers believe that JAZ proteins are the target protein of COI1 and the degradation of JAZ proteins is a key step to relieve the inhibition of the JAs pathway. However, in 2010, Sheard et al. proposed different views on JAs receptors through the analysis of crystal structure and confirmed that the COI1–JAZ complex is a high-affinity receptor for the bioactive JA-Ile; that is, COI1 and JAZ are coreceptors of JA signaling [41]. It is currently believed that plants perceive stimuli from the external environment to generate JA-Ile, which promotes the interaction between COI1 and JAZ proteins. Subsequently, JAZ proteins are degraded after being transferred to the 26S proteasome, and simultaneously, transcription factors (TFs) are released to activate the expression of downstream genes (Figure 2).

### 4.2. Jasmonic Acid Signal-Regulated Transcription Factor

JA-Ile activates the MYC transcription factors by directly binding to JAZ and COI1, which results in the degradation of JAZ through the 26S proteasome pathway (Figure 2). Recent studies have shown that the MYB transcription factors also bind with JAZ repressors and can be activated by the degradation of JAZ in the presence of JA-Ile. In addition, several other transcription factors (TFs) such as NAC, ERF, and WRKY are also involved in the JA signaling. These JA-responsive TFs regulate the expression of many genes involved in the growth and development of plants, and especially the responses and adaptation of plants to the environment (Figure 3). Studies have also shown that JA signaling can also induce the MAP kinase cascade pathway [42], calcium channel [43], and many processes that interact with signaling molecules such as ethylene, salicylic acid, and abscisic acid to regulate plant growth and development [44].

#### 4.2.1. MYC Transcription Factor

The basic helix–loop–helix (bHLH) transcription factor MYC2 is a well-known regulatory protein encoded by the *JIN1* gene. Most members of the JAZ protein family interact with MYC2 [45]. For a long time, it was believed that only the MYC2 protein can directly interact with the JAZ protein. In 2011, Fernandez-Calvo et al. identified that two other bHLH proteins, MYC3 and MYC4, share high sequence similarity with MYC2, suggesting they probably have similar functions. Indeed, MYC3 and MYC4 interact with JAZ proteins in vivo and in vitro, have similar DNA-binding specificity to MYC2, and act synergistically and distinctly with MYC2 [46]. A closely related TF, MYC5 (bHLH28), is induced by JAs and required for male fertility [47]. Besides transcriptional activators, JA-associated MYC2-like (JAM) proteins, JAM1, JAM2, and JAM3, were discovered as transcriptional repressors via forming protein–protein interactions with JAZs to regulate JAs responses [48].

#### 4.2.2. MYB Transcription Factor

Most of the JAs-responsive MYB TFs belong to the R2R3-MYB family, which are widely distributed in plants and required for many processes. Schmiesing et al. showed that MYB51 and MYB34 regulate the synthesis of tryptophan and glucosinolates and act downstream of MYC2 [45]. However, many studies have found that MYB TFs can directly bind to JAZ proteins, indicating the release from JAZs to activate their target genes. For instance, in *Arabidopsis*, MYB21 and MYB24 are key factors in stamen and pollen maturation [49], and MYB75 can positively regulate the anthocyanin accumulation and trichome initiation [50]. Recently, a set of MYB TFs, MYB11, MYB13, MYB14, MYB15, and MYB16, were identified as repressors in the regulation of rutin biosynthesis in tartary buckwheat [51,52].

#### 4.2.3. NAC Transcription Factor

ATAF1 and ATAF2 TFs in the *Arabidopsis* NAC family are both induced by JA signaling and involved in plant resistance to drought, salt stress, *Botrytis cinerea*, and other pathogens [53]. At the same time, ATAF1 and ATAF2 have an important regulatory effect on oxidative stress, flowering, and pod development of plants [54]. Two other NAC TFs in *Arabidopsis*, ANAC019 and ANAC055, are also present downstream of the MYC2 protein and regulate seed germination, cell division, and the synthesis of secondary walls of cells [55]. In addition, ATAF1, ATAF2, ANAC019, and ANAC055 are also involved in the crosstalk between JA and SA signaling pathways [53,54,55].

#### 4.2.4. Ethylene-Responsive Factor (ERF) Transcription Factor

Microarray experiments at the genetic level have confirmed that JA signaling can induce the transcription of many *ERF* genes. The first evidence for a link between AP2/ERF TFs and JA signaling was found in *Catharanthus roseus*. The JAs-induced ORCA proteins, ORCA2 and ORCA3, belong to the AP2/ERF-domain family and can activate the expression of monoterpenoid indole alkaloid biosynthesis genes [56]. Based on the observation of ORCAs, the *Arabidopsis* ERF proteins, ERF1 and ORA59, function dependently on JAs and/or ET for the defenses against *Botrytis cinerea* [57,58]. Moreover, ORA59, rather than ERF1, acts as the integrator of JAs and ET signals [58] and regulates the biosynthesis of hydroxycinnamic acid amides [59]. The JAs-induced ORA47 can activate the expression of the JAs biosynthesis gene *AOC2*, indicating that ORA47 might act as an important regulator in the positive JAs-responsive feedback loop [60]. Moreover, JAs-responsive AtERF3 and AtERF4 act as repressors by not only down regulating their target genes’ expression, but also interfering with the activity of other activators [61]. Interestingly, the activity of above TFs is not directly repressed by JAZ proteins, suggesting the presence of adaptors or corepressors in the JA signaling pathway.

#### 4.2.5. WRKY Transcription Factor

WRKY transcription factors play an important regulatory role in plant development, senescence, and coping with environmental stress. In *Arabidopsis*, there are 89 members in the WRKY transcription factor family. It has been shown that some WRKY TFs are regulated by the JA signaling pathway, such as WRKY70 [62], WRKY22 [63], WRKY50 [64], WRKY57 [65], and WRKY89 [66]. These WRKY transcription factors are mostly associated with plant defense functions. In *Nicotiana attenuata*, two WRKY transcription factors, NaWRKY3 and NaWRKY6, regulate the expression of JAs biosynthesis-related genes (*LOX*, *AOS*, *AOC*, and *OPR*) to increase the levels of JA and JA-Ile [67]. In addition, *Arabidopsis* WRKY57 interacts with the inhibitor JAZ4/JAZ8 in the JA signaling pathway and the inhibitor IAA29 in the auxin signaling pathway, thereby regulating the interaction between JA and auxin-mediated signaling pathways and effects on plant leaf senescence [65].

## 5. Biological Processes Involved in Jasmonic Acid Signaling

### 5.1. Environmental Responses Affected by Jasmonic Acid Signaling

JA and its derivatives are plant signaling molecules closely related to plant defense and resistance to microbial pathogens, herbivorous insects, wounding, drought, salt stress, and low temperature. In addition to the traumatic signals, oligosaccharide signals induced by pathogens and fungal elicitors were also found in tomatoes, which ultimately activated the JA signaling pathway. Upon mechanical wounding in tomato, the prosystemin is hydrolyzed into a systemin, which can be transported to other cells via the apoplast and interacts with the cell surface receptor SR160 (a protein rich in leucine repeat units) to finally activate JAs responses. In this section, we will discuss the role of JA signaling in regulating plant responses in varying environments.

#### 5.1.1. Effect of Light on Jasmonic Acid Signal Changes

To a large extent, the early development of plants is affected by light. It is observed that JA signaling mediates two aspects of the plant response to light: the photomorphogenesis of plants and the damage of plants by UVB. Red light/far-red light-mediated photomorphogenesis was observed in *Arabidopsis* and rice to be affected by JA signaling [68]. The involvement of the JA signaling pathway was also observed in blue light-mediated light morphogenesis in *Arabidopsis* and tomato [69,70]. Enhancement of UVB radiation induces the biosynthesis of JAs in the *Nicotiana* and *Brassica* genera to initiate the JA signaling pathway, and these processes are associated with chemical defenses of plants [68,71].

#### 5.1.2. Effect of Temperature on Jasmonic Acid Signal Changes

The JA signaling pathway is involved in the response and adaptation process of plants to low temperatures. Recent studies on bananas have shown that the MeJA treatment can significantly induce the expression of MYC-family TFs and many cold-responsive genes (*MaCBF1*, *MaCBF2*, *MaCOR1*, *MaKIN2*, *MaRD2*, *MaRD5*, etc.) after cold storage, thereby reducing the damage of plants caused by cold [72]. It was also shown that the endogenous JA content of the banana (*Musa acuminata*) decreased slightly after low-temperature treatment, and the change was not significant [72]. However, in the cold, the MYC gene is activated rapidly in response to exogenous MeJA and resynthesizes a large amount of JAs in the plant body to protect against cold damage. In addition, studies in tomato (*L. esculentum*) [73], pomegranate (*Punica granatum*) [74], loquat (*Eribotrya japonica*) [75], mango (*Mangifera indica*) [76], and guava (*Psidium guajava*) [77] have shown that exogenous MeJA treatment can induce heat shock protein family transcription, increase antioxidant synthesis, reduce lipoxygenase activity, and increase plant resistance to cold damage (with 0 °C upper temperature). At present, there are relatively few reports on the JA signal involved in plant heat resistance, but JA is a key signal molecule for the formation of sesquiterpene induced by heat shock in eaglewood (*Aquilaria sinensis*) [1,2,78].

#### 5.1.3. Effect of Drought on Jasmonic Acid Signal Changes

Many studies have found that JA signaling pathways are involved in drought stress. In *Arabidopsis thaliana* [79] and citrus (*Citrus paradisi* × *Poncirus trifoliate*) [80], it was found that the increase of endogenous JA content after drought stress was rapid and transient, and then gradually decreased to the basal level with the prolonged stress. On the other hand, the application of exogenous JA can also effectively alleviate the damage caused by drought to plants. Peanut (*Arachis hypogaea*) seedlings treated with MeJA showed enhanced drought resistance [81]. It was also observed that MeJA treatment can improve drought resistance in rice (*Oryza sativa*) [82], soybean (*Glycine max*) [83], and broccoli (*Brassica oleracea*) [84] by adjusting metabolism. Studies on broad bean (*Vicia faba*) and barley (*Hordeum vulgare*) have shown that MeJA may regulate stomatal closure through K^+^ channels, thereby increasing the ability of plants to resist drought [85,86].

#### 5.1.4. Effect of Salt on Jasmonic Acid Signal Changes

It was found that the endogenous JA content increased significantly after salt treatment in *Arabidopsis thaliana* [87], tomato (*L. esculentum*) [88], and potato (*Solanum tuberosum*) [89], among other plants. Moreover, JA content increased rapidly and persistently in salt-sensitive plants, while changes in JA content in salt-tolerant plants were not significant [89]. It has also been observed that exogenous JA can enhance the resistance of plants such as pepper (*Capsicum annuum*) [90] and verbena (*Rupestris riparia*) [91] to salt stress.

#### 5.1.5. Effect of CO_2_ Concentration on Jasmonic Acid Signal Changes

Ballhorn et al. found that high concentrations of CO_2_ (500, 700, and 1000 μmol/mol) led to an increase in the release of JAs (MeJA and cis-JA) into the environment in lima bean (*P. lunatus*) [92]. The induced defense of plants infested with nematodes may be affected by elevated CO_2_ concentrations, and CO_2_-induced changes in plant resistance may result in genotypic-specific responses of plants to nematodes under elevated CO_2_ conditions [93]. However, there are few studies on the JA signaling pathway in plants under the action of CO_2_.

#### 5.1.6. Effect of Ozone on Jasmonic Acid Signal Changes

The endogenous JA content of wild-type *Arabidopsis* increased significantly after ozone treatment. Experiments with the ozone-sensitive mutants *rcd1* (radical-induced cell death 1) [94] and *oji1* (ozone-sensitive and jasmonate-insensitive) [91,95] and the JA signal mutant *jar1* [96] have shown that exogenous MeJA can inhibit the spread of programmed cell death caused by ozone. Repression of the JA signaling pathway causes the plant to have a more intense response to ozone. Application of exogenous MeJA also resulted in reduced sensitivity of the hybrid poplar (*Populus maximowizii* × *P. trichocarpa*) [97] and tomato (*L. esculentum*) [96] to ozone. However, recent studies on cotton (*Gossypium hirsutum*) have shown that MeJA exhibits inhibition of ozone damage diffusion only at high concentrations (volume fraction: 685) of ozone, accompanied by antagonism with ethylene [98].

### 5.2. Gene Chip and Proteomics Studies on Jasmonic Acid Signal Changes

RNA-seq and proteomics studies further confirm the involvement of the JA signaling pathway in regulating the physiological processes of plants. Systematic biological approaches such as using gene chip and proteomics can study various physiological processes of plants at the whole genome level, and thus analyze the relationship between various metabolic pathways. Jung et al. identified 137 genes with altered expression levels in MeJA-treated *Arabidopsis* using gene chip technology. Among them, 74 genes were upregulated, including JAs biosynthesis genes, various defense genes (such as *pdf1.2*, encoding genes of myrosinase-binding protein), oxidative stress genes (oxidases, glutathione transfer), aging-related genes, cell wall modification-related genes, hormone metabolism-related (such as ACC oxidase, responsible for ethylene synthesis) genes, and genes involved in storage, signal transduction, and primary and secondary metabolism. The 63 downregulated genes included photosynthesis-related genes (Rubisco enzyme gene, chlorophyll protein gene, early photoinduced protein gene), cold regulatory genes, drought-responsive genes, defense-responsive genes, plant growth and development-related genes, cell wall-modification -related genes, and some other unknown functional genes, etc. [99]. Chen et al. compared the protein content of *Arabidopsis thaliana* before and after MeJA treatment by proteomics and found 186 differentially expressed proteins. These proteins are involved in plant photosynthesis, carbohydrate metabolism, hormone metabolism, secondary metabolism, product transport, stress and defense, and gene transcription [100].

## 6. Future Prospects

In the last decades, the JA signaling pathway has been studied extensively, but our knowledge about the role of JA signaling in response to different environmental stimuli is limited. Environmental signals usually result in a complex response network regulated by multiple signaling pathways. The mechanism of action of JA signaling in plant–environment interaction is still not clear. A series of signal transduction pathways related to JAs biosynthesis and transmission are well known, but it has not been systematically studied how different environmental signals are perceived by plants and initiate JAs synthesis. Due to the large number of receptors and kinases on the cell membrane, different biotic and abiotic signals stimulate the activation of different enzymes, accompanied by a series of complex reactions, such as calcium channel and potassium channel opening. Therefore, the perception of environmental signals has still a lot of research space. The studies on JAs receptors has made great progress, and the JA signal transduction pathway has also been established, but there are still many questions regarding the regulatory process which need to be answered.

## Figures and Tables

**Figure 1 ijms-20-02479-f001:**
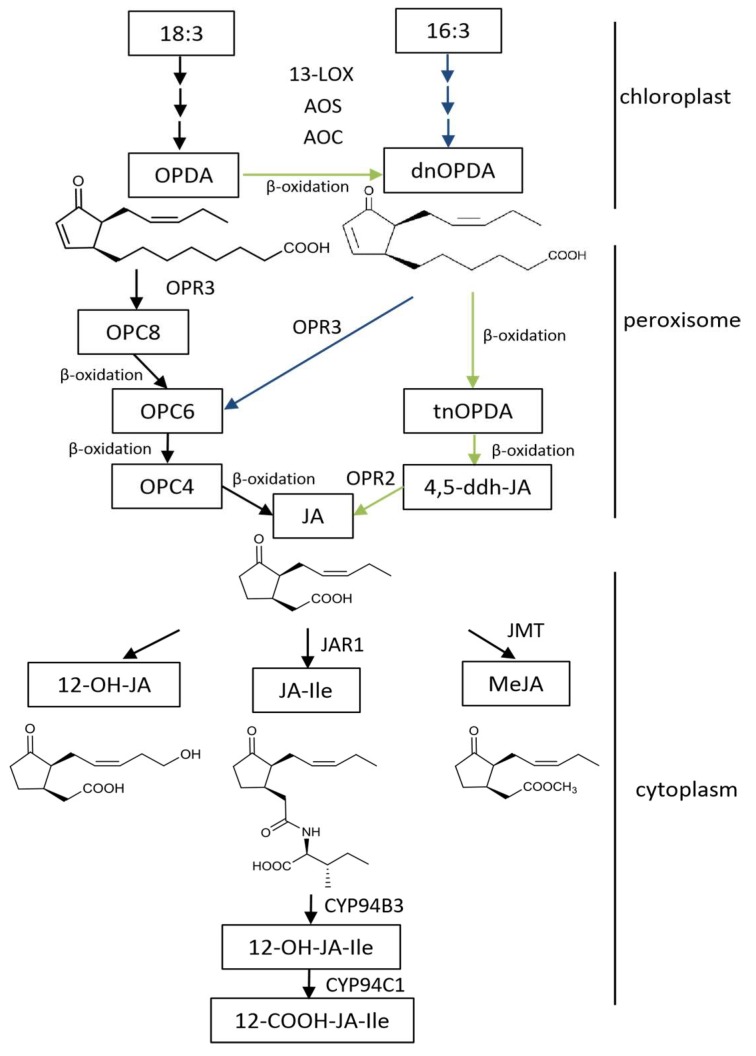
Scheme of the JAs biosynthesis pathway in *Arabidopsis thaliana*. The enzymes and the intermediates are indicated as follows: LOX for lipoxygenase, AOS for allene oxide synthase, AOC for allene oxide cyclase, OPR3 for OPDA reductase, JAR1 for jasmonate resistant 1, JMT for JA carboxyl methyltransferase; 18:3 for α-linolenic acid, 16:3 for hexadecatrienoic acid, OPDA for 12-oxo-phytodienoic acid, dnOPDA for dinor-12-oxo-phytodienoic acid, OPC8 for 8-(3-oxo-2-(pent-2-enyl)cyclopentyl) octanoic acid, OPC6 for 6-(3-oxo-2-(pent-2-enyl)cyclopentyl) hexanoic acid, OPC4 for 4-(3-oxo-2-(pent-2-enyl)cyclopentyl) butanoic acid, tnOPDA for tetranor-OPDA, 4,5-ddh-JA for 4,5-didehydrjasmonate, JA for jasmonic acid, JA-Ile for jasmonoyl-L-isoleucine, and MeJA for methyl jasmonate.

**Figure 2 ijms-20-02479-f002:**
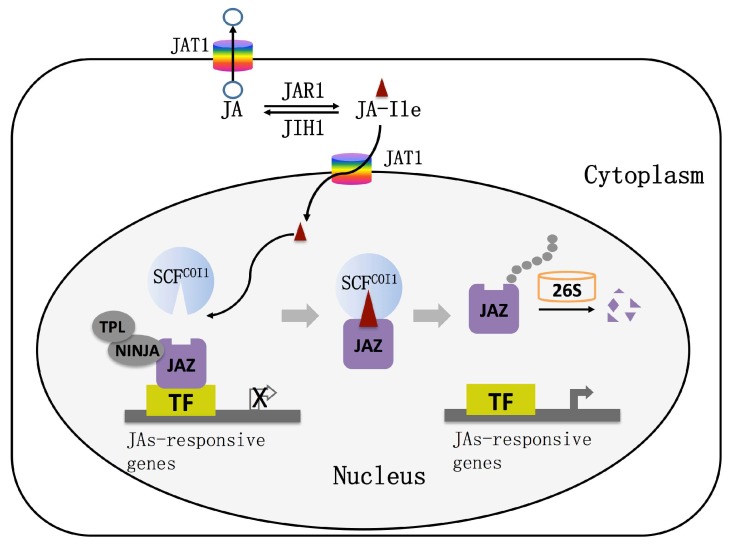
The working model of jasmonic acid transport and signaling pathway. JAT1: jasmonic acid transfer protein1; SCF: Skp1, Cullin and F-box proteins; COI1: coronatine insensitive1; JAZ: jasmonate ZIM-domain protein; TF: transcription factor; TPL: TOPLESS protein; NINJA: NOVEL INTERACTOR OF JAZ; 26S: 26S proteasome.

**Figure 3 ijms-20-02479-f003:**
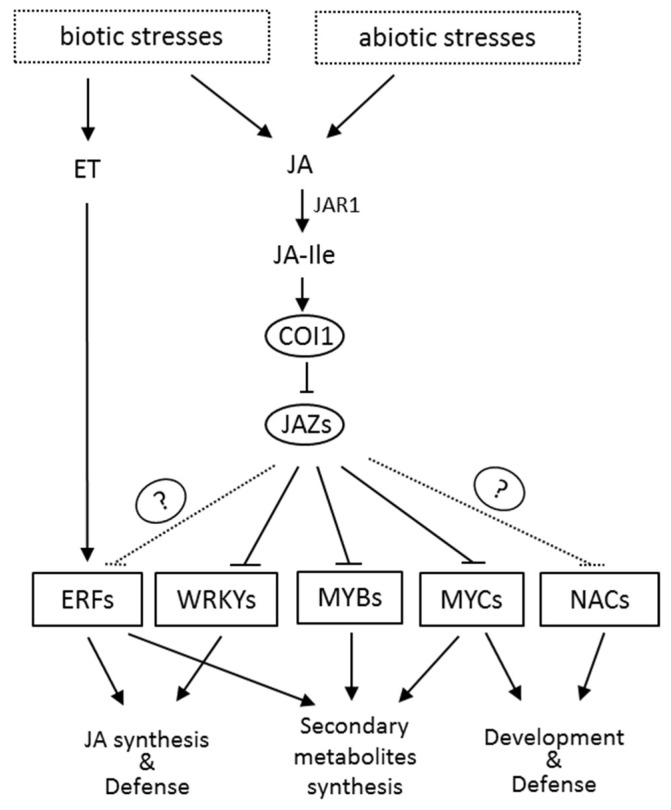
The regulation network of the jasmonic acid signaling pathway. Biotic and abiotic stresses induce the synthesis of JA, which can be converted to the biologically active JA-Ile by JAR1. Perception of JA-Ile by its receptor COI1 triggers the degradation of JAZ repressors, leading to the release of downstream transcription factors and the regulation of JAs-responsive genes in various processes. The question marker indicates an adaptor protein which is not identified yet.

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
