# Peer review of "Jasmonic Acid Signaling Pathway in Plants"

_ijms, 2019, doi:10.3390/ijms20102479_

Reviewer 1 Report

I recommend to write a review focused on chapter 5: Biological  (and ecological) processes involved in jasmonic acid signaling. I would focus the writing on non-model plants, for which literature reviews may be needed. Please, pay special attention to the English language and editing.

Other chapters have been often reviewed.

Author Response

Maybe you're right. I'm also interested in this topic. However, editor-in-chief of this magazine asked us to write a review of plant jasmonic acid signaling pathways. We have requested English-speaking professionals to help us polish the language of this review.

There are many reviews on the biosynthesis of jasmonic acid. In this review, we have not discussed much about this aspect.

Reviewer 2 Report

The review "Jasmonic Acid Signal Pathway in Plants" is a timely contribution to the discussion and understanding of the role and importance of Jasmonates in plants.

 The introduction, 34-47, presents the topic in relation to biotic stress. Then it states it will discuss response to environmental stimuli. I would prefer the authors introduce earlier the response to environment and add also "abiotic stress"

it can mediate 42 responses against abiotic stresses, induce a series of gene expression regulate plant stress responses (add: "and" regulate...)

In tomato, Pearce et 53 al (1991) found a systemin in tomato hydrolyzed into (a) systemin,

81 enzymes in the synthetic pathway haven (+been) identified,

89 takes place in the chloroplast, which is then (the) converted to

90 modification of jasmonic acid "is occured" in the cytoplasm (is occurring / occurs)

95 stressesThe  (+dot and space)

fig. 1 wound /not "would"

in the legend, explain the acronyms (JA2L, PR-STH, TD, C3)

JA and JA-Ile induced 113 by systemin act as signals and are transported to adjacent sites...

systemin play a direct role during the process of 119 long-distance signal transmission and "are" a mobile (singular)

the sieve in the phloem 129 molecules "have" (change "molecules") 

[26]. recently / Recently

confirmed by (the) Larrieu et al [28].

134 The/An experiment found that

demonstrated that even (+when) 137 vascular bundle transmission was blocked, there was a rapid

158 12 JAZ proteins in Arabidopsis, both/all of which have.... Jas

159 and ZIM [39]. This ref is: 

Growth of Arabidopsis seedlings on high fungal doses of Piriformospora indica has little effect on plant performance, stress, and defense gene expression in spite of elevated jasmonic acid and jasmonic acid-isoleucine levels in the roots.

Maybe you want to quote instead: 

A critical role of two positively charged amino acids in the Jas motif of Arabidopsis JAZ proteins in mediating coronatine- and jasmonoyl isoleucine-dependent interactions with the COI1 F-box protein. Melotto M, Mecey C, Niu Y, Chung HS, Katsir L, Yao J, Zeng W, Thines B, Staswick P, Browse J, Howe GA, He SY. Plant J. 2008; 55(6):979-88.

Figure 2. Jasmonic acid transport protein working model (not only transport, also signaling)

MAP cascade pathway [42]: MAP kinase

MYB TFs: activates in  (are activated) 216 a similar way like MYC2 through degradation of JAZ proteins,

165 believed that plant perceive stimuli /plants perceive or plant perceives... to generat! 

167 26s proteasome: 26S

ERF4 is mutually inhibited from EFR1 and EFR2 [53, 54]. :ERF

In tobacco, two WRKY 248 transcription factors, NaWRKY3 and NaWRKY6, it would be better to state: in Nicotiana attenuata (NaWRKY)

check and proof edit

542 63. Skibbe M, Qu N, Galis I, Balawin I T. Induced plant defenses in the natural environment: Nieotiana 543 attenuala: Nicotiana attenuata

regulating the interaction between JA and Auxin 253 mediated signaling pathways and effects plant leaf senescence (effects on plant.../ or: affect plant)

Figure 3. Provide a better readability of the boxes (pathoge?); SiNAC: which Solanum species?; SNAC1?; dependent against...: please adjust, and improve understandability)

258 and its derivativesare... Add space

281 It was also showed (shown)

313 Ballhorn

5,2 Transcriptomic- gene chip and proteomic studies: this article, already discussed as 85., may be quoted also here in this paragraph

85. Zhang H, Zhang Q, Zhai H, Li Y, Wang X, LiuQ, He S. Transcript profile analysis reveals important 603 roles of jasmonic acid signalling pathway in the response of sweet potato to salt stress. Sci Reports, 2017, 7:40819.

I would like you to quote the two following articles, in the paragraph 5.1.4 in relation to salt stress response

Oxylipin dynamics in Medicago truncatula in response to salt and wounding stresses.

De Domenico S, Taurino M, Gallo A, Poltronieri P, Pastor V, Flors V, Santino A.

Physiol Plant. 2019;165(2):198-208. doi: 10.1111/ppl.12810.

and

The salt-responsive transcriptome of chickpea roots and nodules via deepSuperSAGE.

Molina C, Zaman-Allah M, Khan F, Fatnassi N, Horres R, Rotter B, Steinhauer D, Amenc L, Drevon JJ, Winter P, Kahl G.

BMC Plant Biol. 2011;11:31. doi: 10.1186/1471-2229-11-31.

Author Response

The introduction, 34-47, presents the topic in relation to biotic stress. Then it states it will discuss response to environmental stimuli. I would prefer the authors introduce earlier the response to environment and add also "abiotic stress"

The introduction has been revised as required.

it can mediate 42 responses against abiotic stresses, induce a series of gene expression regulate plant stress responses (add: "and" regulate...)

“and” was added before “regulate” .

In tomato, Pearce et 53 al (1991) found a systemin in tomato hydrolyzed into (a) systemin,

This sentence has been revised as required.

81 enzymes in the synthetic pathway haven (+been) identified,

“been” was added before “identified” .

89 takes place in the chloroplast, which is then (the) converted to

“the” was deleted.

90 modification of jasmonic acid "is occured" in the cytoplasm (is occurring / occurs)

“is occured” was changed into “occurs” .

95 stressesThe  (+dot and space)

Dot and space have been added to “stressesThe”

fig. 1 wound /not "would"

“would” was changed into “wound” in Fig. 1.

in the legend, explain the acronyms (JA2L, PR-STH, TD, C3)

JA2L, PR-STH, TD and C3 have been explained in the legend.

JA and JA-Ile induced 113 by systemin act as signals and are transported to adjacent sites...

The sentence has been revised.

systemin play a direct role during the process of 119 long-distance signal transmission and "are" a mobile (singular)

“are” was changed into “is” .

the sieve in the phloem 129 molecules "have" (change "molecules")

“molecules” was added after “sieve” .

 [26]. recently / Recently

“recently” was changed into “Recently” .

confirmed by (the) Larrieu et al [28].

“the” was deleted.

134 The/An experiment found that

“The” was changed into “An”.

demonstrated that even (+when) 137 vascular bundle transmission was blocked, there was a rapid

“when” was added before “vascular” .

158 12 JAZ proteins in Arabidopsis, both/all of which have.... Jas

“both” was changed into “all”.

159 and ZIM [39]. This ref is:

Growth of Arabidopsis seedlings on high fungal doses of Piriformospora indica has little effect on plant performance, stress, and defense gene expression in spite of elevated jasmonic acid and jasmonic acid-isoleucine levels in the roots.

Maybe you want to quote instead:

A critical role of two positively charged amino acids in the Jas motif of Arabidopsis JAZ proteins in mediating coronatine- and jasmonoyl isoleucine-dependent interactions with the COI1 F-box protein. Melotto M, Mecey C, Niu Y, Chung HS, Katsir L, Yao J, Zeng W, Thines B, Staswick P, Browse J, Howe GA, He SY. Plant J. 2008; 55(6):979-88.

This ref has been replaced.

Figure 2. Jasmonic acid transport protein working model (not only transport, also signaling)

“and signaling” was added after “transport” .

MAP cascade pathway [42]: MAP kinase

“kinase” was added after “MAP” .

MYB TFs: activates in (are activated) 216 a similar way like MYC2 through degradation of JAZ proteins,

“activates” was changed into “are activated”.

165 believed that plant perceive stimuli /plants perceive or plant perceives... to generat!

This sentence has been revised as required.

167 26s proteasome: 26S

“26s” was changed into “26S”.

ERF4 is mutually inhibited from EFR1 and EFR2 [53, 54]. : ERF

“ERF4” was changed into “ERF”.

In tobacco, two WRKY 248 transcription factors, NaWRKY3 and NaWRKY6, it would be better to state: in Nicotiana attenuata (NaWRKY)

“tobacco” was changed into “Nicotiana attenuata”.

check and proof edit

 542 63. Skibbe M, Qu N, Galis I, Balawin I T. Induced plant defenses in the natural environment: Nieotiana 543 attenuala: Nicotiana attenuata

“Nieotiana attenuata” was changed into “Nicotiana attenuata”.

regulating the interaction between JA and Auxin 253 mediated signaling pathways and effects plant leaf senescence (effects on plant.../ or: affect plant)

“on” was added after “effects” .

Figure 3. Provide a better readability of the boxes (pathoge?); SiNAC: which Solanum species?; SNAC1?; dependent against...: please adjust, and improve understandability)

Figure 3 has been adjusted as required.

258 and its derivativesare... Add space

“derivativesare” was changed into “derivatives are”.

281 It was also showed (shown)

“showed” was changed into “shown”.

313 Ballhorn

Ballhorn is right in here.

5,2 Transcriptomic- gene chip and proteomic studies: this article, already discussed as 85., may be quoted also here in this paragraph

The reviewer's opinion is correct.

85. Zhang H, Zhang Q, Zhai H, Li Y, Wang X, LiuQ, He S. Transcript profile analysis reveals important 603 roles of jasmonic acid signalling pathway in the response of sweet potato to salt stress. Sci Reports, 2017, 7:40819.

Reference 85 and 86 have been replaced on the basis of reviewers' opinions.

I would like you to quote the two following articles, in the paragraph 5.1.4 in relation to salt stress response

 Oxylipin dynamics in Medicago truncatula in response to salt and wounding stresses.

 De Domenico S, Taurino M, Gallo A, Poltronieri P, Pastor V, Flors V, Santino A.

 Physiol Plant. 2019;165(2):198-208. doi: 10.1111/ppl.12810.

 and

 The salt-responsive transcriptome of chickpea roots and nodules via deepSuperSAGE.

 Molina C, Zaman-Allah M, Khan F, Fatnassi N, Horres R, Rotter B, Steinhauer D, Amenc L, Drevon JJ, Winter P, Kahl G.

 BMC Plant Biol. 2011;11:31. doi: 10.1186/1471-2229-11-31.

Round  2

Reviewer 1 Report

Please consider suggestions annotated in the pdf.

Author Response

JA-Ile is a JA derivative, an AA-conjugate, like several other JA-AA conjugates. NOT a complex; NOT genereting other complexes. Complexes have been replaced by derivatives. are included in "environmental signal perception" "environmental signal perception" have been revised. maybe better in plural? Plant has been replaced by plants. define at first use, line before Jasmonic acid has been replaced by JA. italics, check throughout the text (cis, trans, gene names, mutants, etc) italics, cis, trans, gene names, mutants, etc have been check throughout the text. look the way you named it at the abstract! Jasmonic acid has been replaced by JA. whom? JA?? This has been modified. not true! JA-Ile, the active jasmonate actually induces the opening. This is explode by p. syringae to facilitate plant colonization thanks to the JA-Ile mimic Coronatine. This part has been revised. one or two?? keep one This has been modified. three: Nature Chemical Biology: doi:10.1038/nchembio.2540 two has been replaced by three. alpha-linolenic... alpha has been added before linolenic. for JA not needed This part has been revised. you did that before This part has been revised. tuberonic acid too: www.plantphysiol.org/cgi/doi/10.1104/pp.110.168617 also have a look here: http://dx.doi.org/10.1016/j.nbt.2015.11.001 This has been modified. plural Plant has been replaced by plants. have a look at: JA but not JA-Ile is the cell-nonautonomous signal activating JA mediated systemic defenses to herbivory in Nicotiana attenuata This part has been revised. and chini et al. Vol 448|9 August 2007| doi:10.1038/nature06006 This has been modified. there are 13 JAZs 12 has been replaced by 13. most of them, JAZ7/8/13 don't. NO jas domain This part has been revised. reference for this Reference has been added. no need to re-define names you already did. Check all the text for this Abbreviation has been check throughout the text. knockout mutant? NO MYC2? clarify This part has been corrected. and are...? This part has been revised. don't get it. ERF4? what ERF? ERF has been annotated in this article. COI1 This part has been revised. MeJA treatment.... cause the signaling molecule is surely JA-Ile This part has been revised. maybe better focuss on RNAseq... more modern Microarray has been replaced by RNAseq. in the... the has been added before last. I would say just little... This part has been revised.
